# Promote or Crowd Out? The Impact of Environmental Information Disclosure Methods on Enterprise Value

Anrong Gao [1], Tianren Xiong [1], Yuxi Luo [1,2,*] and Defeng Meng [1,2,*]

1 School of Economic & Management, Guangxi Normal University, Guilin 541006, China
2 Pearl River-Xijiang River Economic Belt Development Institute, Guangxi Normal University, Guilin 541004, China
* Correspondence: yluogxnu@mailbox.gxnu.edu.cn (Y.L.); mdf_gxnu@mailbox.gxnu.edu.cn (D.M.)

**Abstract:** Environmental information disclosure is a concrete practice for enterprises to actively implement the concept of green and sustainable development, which has great significance for enterprises to gain long-term competitive advantages. The academic world has widely discussed the relationship between environmental information disclosure and the economic performance of enterprises, but how the heterogeneity of environmental information disclosure methods affects the enterprise value has not been explored. This paper aims to answer two questions: (1) what is the impact of Ecomark and ESG on enterprise value? and (2) how does the interaction between Ecomark and ESG influence enterprise value? Utilizing the listed Japanese electrical equipment manufacturing enterprises dataset from 2008 to 2021, we employed the fixed panel linear regression model to confirm the relationship between Ecomark and ESG in enterprise value, and further used a moderating effect model to verify the existence of the crowd-out effect of ESG performance on Ecomark through enterprise value. In addition, a robustness check scheme was designed and performed to test the model settings, outliers and endogeneity issues. The main findings show that the obtaining of Ecomark certification and good ESG performance can help to improve enterprise value, but they may be altered regarding the heterogeneity of environmental information disclosure methods, further causing differences in enterprises' time and economic cost burdens. Such differences increase the attractiveness of ESGs to investors, thereby crowding out the impact of Ecomark on enterprise value. Our conclusion reveals the mechanism of the heterogeneity of environmental information disclosure methods towards enterprise value, which offers a valuable reference for investors to evaluate enterprise value and paves the way for enterprise decision-makers and authorities to optimize their environmental information disclosure.

**Keywords:** Japan; enterprise value; Ecomark; ESG; product life cycle

## 1. Introduction

Rapid economic development, technological progress, unsustainable consumption and a growing global population mean that we can no longer avoid negative impacts on the environment. This has led to the major environmental degradation and ecological disasters facing humanity today. Drastic changes in climate, water and air pollution, damage to flora and fauna, the depletion of the ozone layer, acid rain and deforestation are among the most common environmental disasters and events. With the deepening of research, people's attention to the above problems has increased dramatically. According to the results of the "Public Opinion Survey on Global Warming Countermeasures" conducted by the Cabinet Office of Japan in 2016, 87.2% of Japanese people are concerned about the present and future global environmental problems [1]. People are not only concerned about the Earth's environmental problems but also take practical environmental protection actions to alleviate environmental problems. The proportion of Japanese people choosing energy-saving and high-efficiency products when replacing household appliances increased

from 46.1% in 2007 to 56.3% in 2016. With the continuous improvement of consumers' awareness of environmental protection, enterprises who proactively participate in third-party environmental protection measures continue to increase [2,3]. The proportion of Japanese listed companies obtaining "ISO14001"certification was constantly maintained at 68.8% to 75.9% from 2007 to 2019 [4]. Such practices help enterprises to fulfill their social responsibilities and maximize their value. Additionally, Japanese enterprises release information on environmental management through environmental accounting systems and environmental impact statements to convince investors and consumers to recognize their efforts in environmental protection. In recent years, obtaining Ecomark system certification and releasing ESG performance reports have been commonly used to disclose enterprises' environmental information. Darnall et al. (2022) pointed out that Japanese listed companies who adopted the ESG method tended to disclose 39% more environmental information than those who released their environmental information traditionally [5]. Thus, a growing number of Japanese enterprises are following this procedure.

Among them, the Ecomark system is an environmentally friendly certification system to evaluate whether a product's manufacturing process and its entire life cycle meet the environment protection standards. It is a consumer-oriented environmental measure to promote the purchase of household environmental protection products and enterprises' green procurement, thus indirectly improving investors' investment intention in related enterprises. In contrast, the ESG system means that enterprises and investors integrate environmental, social and corporate governance issues into their business models. The ESG system represents enterprises' green development strategy and governance mechanism, which is more sensitive to the capital market and in line with the investors' intentions. With the popularity of the global green development concept, more and more enterprises attach importance to their own ESG rating and performance. The academic world often focuses on how the ESG rating affects value and corporate strategies (Cappucci, 2018) [6]. The Financial Services Agency (FSA) of Japan and the Tokyo Stock Exchange integrated the ESG system in 2015, requiring enterprises to pay specific attention to the importance of environmental, social, governance and other non-financial performance aspects in accordance with enterprises' purposes of attracting investment, promoting sustainable development and fulfilling social responsibilities. Miles and Covin (2000) showed that enterprises that adopt independent environmental protection measures will be given a higher evaluation by stakeholders and form competitive advantages in products, which may promote their corporate financial performance [7]. Therefore, in order to form a win–win situation regarding economic development and environmental protection, academia and researchers have carried out analyses and research on the relationship between the environmental performance and economic performance of enterprises through various aspects.

Since Porter (1991) and Porter and Linde (1995) [8,9], research on the relationship between corporate environmental performance and corporate economic performance has gradually arisen (Molina-Azor Azorín et al., 2009; Blanco et al., 2009; Fujii et al., 2013; Santos et al., 2019; Ren et al., 2020; Wang et al., 2022) [10–15]. Most of the existing literature has considered the relationship between reducing the environmental load in the production process and enterprises' economic performance. However, reducing the environmental load should not be only limited to products' production stage, but also to their use, consumption, recycling and disposal stages. For instance, utilizing energy-saving technology in the use stage and adopting a recycling-friendly design for electrical products also play an important role in reducing the environmental load. It is necessary to extend the research attention to the whole product life cycle. Although the literature has shown that enterprises' environmental information disclosure through various means may help to improve their economic performance (Nishitani, 2014; Nishitani and Kokubu, 2014; Wang et al., 2021) [16–18], studies considering the whole product life cycle are still lacking.

In this paper, we aim to perform an empirical analysis to reveal the different impacts of Ecomark (from customers' perspectives) and ESG (from investors' and stakeholders'

perspectives) on enterprise value and verify the link between them with consideration of the whole product life cycle.

Given the enriched existing evidence from China, the United States, Australia, Europe and other countries, we found little literature on the linkage between Ecomark and ESG. We thus utilize a sample of Japan's listed electrical equipment manufacturing enterprises to investigate such a link. The main question that we attempt to answer is as follows: how does ESG influence Ecomark regarding enterprise value?

Compared with the existing studies, the main contributions of this paper are as follows: firstly, this paper incorporates the product life cycle theory into the analysis framework of environmental information disclosure in enterprise value and examines the impact of the Ecomark system and ESG performance on enterprise value; secondly, the possible crowd-out effect of ESG performance on Ecomark through enterprise value is investigated.

This paper is organized as follows: Section 2 summarizes the background of Japan's Ecomark system and ESG system; Section 3 reviews the literature on the relationships between Ecomark, ESG and enterprise value and forms research hypotheses; Section 4 defines the data and empirical models; Section 5 discusses the empirical results, robustness check and endogeneity test; and Section 6 concludes with the main findings and remarks.

## 2. A Brief Summary of Japan's Ecomark System and ESG System

### 2.1. Japan's Ecomark System

Due to the increasing awareness of the perniciousness of environmental problems, enterprises have attached various environmental labels to their products to show their environmental protection efforts and environmental performance. Ecomark is one of these environmental labels. The International Organization for Standardization (ISO) has defined environmental labels into three classes: type 1, type 2 and type 3. Each type of environmental label must meet the following requirements: type 1 requires third-party certification, with the highest authority; type 2 is independently formulated by the enterprise according to the performance of the product, and does not require third-party certification; type 3 only quantitatively displays the environmental load information of the product, with no need to determine whether it is environmentally friendly [19]. Ecomark is the only type 1 environmental label in Japan that meets the ISO environmental label classification, which has the following three characteristics: (a) it requires the certification of a third party, rather than any one of the product suppliers or consumers; (b) when formulating standards, one should not only consider the impact of a certain link, such as the production, use and disposal of products on the environment, but also consider whether the environmental load through the whole product's life cycle meets the standards; (c) enterprises should provide a place for stakeholders (product suppliers, consumers, scholars, etc.) to discuss the environmental performance of products.

Japan introduced and implemented the "Ecomark" system in February 1989. Ecomarks are specific graphics printed on products and packaging operated by the Japan Environmental Association, a public welfare consortium. It indicates that a product and its whole life cycle process (production, use, recycling and disposal) involves few or no substances harmful to human health and the environment. Through the promotion of using Ecomarks, enterprises can promote their own investment in environmental improvement and encourage consumers to choose green products. The ultimate goal is to create sustainable social development. According to the regulations of Ecomark issued by the office of the Japan Environmental Association, the basic requirements for domestic and foreign products sold in Japan to obtain an Ecomark are as follows: (a) the environmental load of the products with Ecomark certification (throughout the process of production, use, recycling and disposal) is lower than that of other similar products; (b) through the use of products with Ecomark certification, the environmental load can be reduced, and environmental protection can be expanded [20].

The following example is used to show the general criteria for obtaining Ecomark certification. The case No. 152 Version 1 represents the general criteria for a Japan-made TV

set to obtain Ecomark certification. As shown in Table 1, the environmental load assessment project discusses four major factors: the saving and recycling of resources, the prevention of global warming, the limit and control of the production of harmful substances and the conservation of biodiversity. The life cycle of the product includes the procurement of raw materials, manufacturing, circulation, use and consumption, recycling and disposing. Of these, the items printed with the symbols ◎ and ○ are environmental load assessment items that must meet the standards in order to obtain the Ecomark, and the items not affixed with the symbols ◎ and ○ are items that are not required to meet the standards or are included with other content. A television must meet all of the required certification criteria listed in Table 1 in order to obtain Ecomark certification [21]. For instance, the conservation of biodiversity was considered as a criterion for the obtaining of Ecomark, but it was not specifically discussed as a requirement for the life cycle of television sets—that is, it was not recognized as a requirement for the obtaining of Ecomark for television sets. Some believe that saving energy and raw materials will eventually help to preserve biodiversity.

**Table 1.** Selection of life cycle environmental assessment projects for Japanese televisions.

| Environmental Load Assessment Project | Life Cycle of Product | | | | | |
|---|---|---|---|---|---|---|
| | Procurement of Raw Materials | Manufacturing | Circulation | Use and Consumption | Recycling | Disposing |
| Resource saving and resource recycling | ◎ | | ◎ | | ◎ | ◎ |
| Prevention of global warming | | | | ◎ | | |
| Limit and control the production of harmful substances | | ◎ | | ◎ | ◎ | ◎ |
| Conservation of biodiversity | ○ | | | | | |

Source: No. 152 Version 1 Commentary "TV Version 1" 12 March 2013.

Since the implementation of the Ecomark system, it has been gradually recognized by society. From 2 to 4 March 2015, the Bureau of Ecomark Affairs of the Japan Environmental Association conducted a survey on the awareness of Ecomark. The results showed that 90.7% of the respondents had some knowledge of Ecomark [22]. In addition, according to the progress report of Ecomark in 2015, it can be seen that the number of products with Ecomark certification increased from 265 in February 1989 to 5507 at the end of June 2015, and the number of products with Ecomark certification increased by approximately 21 times [23]. During the same time period, Japan issued the Green Procurement Law (GPL) in April 2001 to promote society's circular and sustainable development, which provides procurement guidance and necessary standards for green procurement practice and promotes the implementation of green procurement from the legal aspect. Ecomark is an important reference for green procurement and has been widely recognized by society. According to the survey results of the Ministry of Environment of Japan (2013), 96.6% of government agencies, enterprises and local public organizations give priority to Ecomark when implementing green procurement [24]. Hayashi and Funabashi (2002) and Loncar et al. (2019) pointed out that the obtaining and use of Ecomark certification can improve the social awareness of environmental protection products, which helps enterprises to promote their positive environmental image in society. In addition, through the implementation of the GPL, green purchases by consumers and producers are being encouraged, so there is a trend towards expanding the market share of products with an Ecomark [25,26]. Some scholarly articles have also concluded that enterprises that have better environmental performance will result in a higher return from the capital market (Yamaguchi, 2008; Takeda and Tomozawa, 2009; and Gupta and Goldar, 2005) [27–29]. Therefore, enterprises that

obtain Ecomark certification may expand the market demand for products and promote their enterprise value.

### 2.2. Japan's ESG System

This paper introduces ESG as an important independent variable. The concept of ESG was first proposed by Goldman Sachs in 2007, aiming to incorporate environmental (E), social (S) and corporate governance (G) factors into investment decisions and evaluate the sustainability of enterprise operation and its impact on social values in the above three dimensions, representing the comprehensive performance of enterprises in the market. In 2014, the FSA first released the Japan Stewardship Code, which encourages investors to improve and promote the corporate value and sustainable growth of their invested companies through consultation and participation. In 2015, the FSA and the Tokyo Stock Exchange issued the first edition of the Japan Corporate Governance Code, which officially started the practice of ESG integration in Japan. Kato (2022) mentioned that the Japan Stewardship Code and the Japan Corporate Governance Code not only affected the enterprises' ESG disclosure strategy but also encouraged enterprises to disclose environmental information under the ESG guidelines [30]. With the rapid development of the ESG policy and regulations, Japan's ESG practices have grown quickly in the past six years. Therefore, studying the relationship between ESG and enterprise value may not only clarify the mechanism between enterprise ESG performance and enterprise value but also be of benefit for promoting the ESG system and improving enterprises' sustainable development.

## 3. Literature Review and Research Hypotheses

### 3.1. Literature Review

In recent years, scholars have carried out much research on the relationship between environmental practices and economic performance. Firstly, researchers used Tobin's Q as a measure to link enterprises' environmental practices and their economic performance. Porter and Linde (1995) claimed that good environmental regulation can strengthen the competitiveness of enterprises and further promote enterprise value [9]. However, the definition of good environmental regulation is not specified. In addition, with the exception of setting a hypothesis, this study did not use specific data and models to further prove the relationship between environmental regulation and enterprise value empirically. Since then, many scholars have carried out relevant empirical analyses and reached different conclusions. Some scholarly papers showed that there is a positive correlation between good environmental practice indicators and Tobin's Q (Dowell et al., 2000; Konar and Cohen, 2001; King and Lenox, 2002; Elsayed and Paton, 2005; Nakao et al., 2010; Nakao et al., 2005; Nishitani and Kokubu, 2012; Farza et al., 2021; and Martha et al., 2022) [31–39]. On the other hand, some research has led to the opposite conclusions, showing that there is either no or a negative significant correlation between environmental practices and enterprises' economic performance (Iwata and Okada, 2011; Rassier and Earnhart, 2010; Hibiki and Managi, 2010; Liu and Zhang, 2016) [40–43]. The above research only focuses on the analysis of whether the environmental practices in enterprises' manufacturing stages have an impact on their economic performance, while not considering how the environmental practices influence enterprise value from other stages within the whole product life cycle. Besides the above, Ruiz-Blanco et al. (2022) pointed out that a high quality of environmental information disclosure leads to better enterprise value and showed the diversity of decision-makers' practical strategies from the aspect of the "greenwashing" phenomenon [44]. Since only little attention has been given to the specific enterprises' environmental practice indicators and their enterprise value in regard to the whole product life cycle view, we introduce Ecomark to bridge such gaps.

Secondly, there is heterogeneity in the research of eco-labels and enterprises' economic outcomes. Tully and Winer (2014) pointed out that the studies on the relationship between the adhesion of relevant labels to environmentally friendly products produced through environmental protection efforts by enterprises in different countries and regions are

different from each other, but most of them are aimed at verifying whether the use of relevant eco-labels on products will affect consumers' choice preferences and thus generate a corresponding premium. At the same time, with the continuous improvement in social attention to environmental issues, enterprises have been strengthening their independent or legal disclosure of environmental information. As a means to disclose enterprises' environmental practice information, the eco-label affects their market value [45]. Nishitani and Kokubu (2014) and Wang et al. (2020) found a positive correlation between the measures of issuing environmental reports and enterprise value, which indicated that the release of environmental reports to publicize the environmental performance of enterprises has a tendency to increase the value of enterprises [18,46]. Brouhle and Khanna (2012) analyzed the actual purchasing data of the Danish paper market and showed that the Nordic Swan eco-label certified products may result in an increase in consumers' purchase propensity [47]. Aguilar and Cai (2010) used conjoint analysis to examine consumers' preferences for wood products in the United States and the United Kingdom and found a price premium for environmental protection certification [48]. Sammer and Wüstenhagen (2006) showed that Swiss-made washing machines with EU energy labels may result in higher consumers' willingness to pay for a more energy-efficient product [49]. Teisl et al. (2002) revealed that enterprises who have the dolphin-safe label on their products tend to increase their shares in the canned tuna market [50]. The above evidence implies that enterprises may benefit from proactively disclosing environmental information, but they insufficiently linked this with Tobin's Q. Although Gao and Nakano (2017) pointed out that eco-labels promote the enterprise value of Japan's listed manufacturing enterprises, there is still very little research on whether the eco-label certification affects Tobin's Q [51].

Thirdly, research conclusions on the impact of ESG on enterprise value have not been reached yet. Most scholars' research results show that ESG has a positive impact on different indicators of enterprise economic performance, and better ESG performance can improve enterprise performance or market valuation. Cajias et al. (2014), Li et al. (2018) and Wong et al. (2021) believe that better ESG performance can promote the growth of enterprise value (Tobin's Q), and the disclosure of ESG information also has a positive impact on corporate profitability (ROA) [52–54]. In addition, some studies also show that the good performance of enterprise ESG has a tendency to promote a stock price increase at the end of the year (Brogi and Lagasio, 2019; Yoon et al., 2018) [55,56]. At the same time, Sila and Cek (2017) studied Australian enterprise data and showed that good ESG performance could enhance enterprise value [57]. Although the ESG's three major dimensions may impact enterprise value via different paths and mechanisms with respect to enterprises' characteristic differences, the overall evaluation of ESG has an enhancing effect on enterprise value (Taliento et al., 2019) [58]. In addition, Sassen et al. (2016) and Landi et al. (2022) pointed out that ESG information disclosure increased investors' capability of risk perception, which can reduce enterprises' total and special risks [59,60]. In the context of special events, enterprises must keep their strategic development goals in line with future environmental and social needs. Thus, the sustainability and quality of environmental information disclosure methods need to be constantly improved (Tettamanzi et al., 2022) [61]. For instance, Takahashi and Yamada (2021) showed that the Japanese listed companies who invested via ESG funds had higher stock returns during the COVID-19 pandemic, but there was no evidence that companies with high ESG scores had higher stock returns [62]. Meanwhile, cases from China demonstrated that ESGs could reduce enterprises' financial risks and significantly enhanced their financial resilience during the COVID-19 pandemic (Broadstock et al., 2021; Zhang and Liu, 2022) [63,64]. Eliwa et al. (2021) and Yue and Wang (2022) also show that the good ESG disclosure of enterprises has a tendency to reduce enterprises' debts and increase their risk tolerance [65,66]. The continuous competitive advantage generated from ESG performance constitutes intangible value, which indirectly leads to the improvement of corporate performance (Ferrero-Ferrero, 2016) [67]. These results indicate that enterprises can broaden financing channels, improve investment efficiency, promote green innovation and indirectly improve enterprise value

by improving their ESG performance. However, some scholars believe that ESG tends to reduce corporate value. Garcia and Orsato (2020) show that in emerging markets, companies are more likely to give priority to capital accumulation, without realizing the potential strategic benefits of socially responsible investment, so ESG will increase corporate costs. Thus, there is a tendency towards the value of enterprises being reduced [68]. Kuo et al. (2021) found a downward trend in airlines' returns on assets in the initial stage of ESG implementation, but this increases over time [69]. Ruan and Liu (2021) also showed that ESG measures of non-state-owned enterprises listed in Shanghai and Shenzhen tend to reduce corporate ROA [70].

It is not difficult to see that the existing literature presents two technical paths for the study of environmental information disclosure and enterprise value. The conclusions on the impact of eco-labels on enterprise value are consistent, but there are still prominent differences regarding the transmission path of ESG enterprise value. Zhong et al. (2022) suggested that the driving effect of ESG on enterprise value is influenced by its own factors and multiple factors such as enterprise scale and profitability. Although the interaction between factors will form different paths, the positive impact on enterprise value has remained consistent [71]. This paper attempts to comprehensively investigate the impact of environmental information disclosure on enterprise value through the interaction of Ecomark and ESG.

*3.2. Research Hypotheses*

According to the signal transmission theory, the disclosure of non-financial information, such as environmental information, by enterprises can lead consumers to identify low-carbon environmental protection products and help enterprises to gain a positive reputation among the public, which may be more acceptable to external stakeholders. With the promotion and popularization of global consumers' awareness of environmental protection, green products will gain brand advantages and become more competitive in the market. The existing literature contains enriched evidence that demonstrates that eco-labels provide more environmental information from a non-financial perspective and can help to reduce the information asymmetry between enterprises and consumers, to affect consumers' preferences and to generate corresponding premiums and improve enterprise value (Brouhle and Khanna, 2012; Aguilar and Cai, 2010; Sammer and Wustenhagen, 2006; Gao and Nakano, 2017) [47–49,51]. Obtaining an Ecomark represents better environmental practices for enterprises, which indicates that an enterprise possesses a perfect environmental governance mechanism and strong social consciousness. Enterprises who have products certified by Ecomark transmit a strong signal to stakeholders that they have great potential regarding sustainable development, which helps them to increase their enterprise value (Nakao et al., 2010; Nakao et al., 2005; Nishitani and Kokubu, 2012) [35–37]. Given the above, we propose Hypothesis 1:

**Hypothesis 1.** *Obtaining Ecomark certification can promote enterprise value.*

With the increase in environmental awareness, the importance of adopting ESG in the enterprise evaluation process is increasing in the financial and capital markets. Sasaki and Hanaeda (2021) suggested that ESG may become the best method of improving enterprises' performance [72]. Enterprises that undertake environmental responsibility and social responsibility can convey a trustworthy signal to stakeholders, which helps enterprises to win the trust and support of stakeholders and to obtain the necessary resources and environment for their sustainable development, as well as to reduce transaction costs and promote enterprises' value creation (Freeman and Evan, 1990) [73]. Lee et al. (2022) showed that passing the ESG performance information actively to consumers can help automobile enterprises to promote their brand and values. Thus, managers should pay more attention to governance and the environment and to the enterprises' long-term sustainable development, instead of emphasizing short-term financial benefits [74]. The improvement of

green and quality management can reduce the waste of resources (Wu et al., 2022) [75]. Enterprises with good ESG performance can achieve higher customer satisfaction (Amel-Zadeh, 2017) [76] and increase the possibility of enterprise value by reducing conflicts between shareholders and management (Nekhili, 2020) [77]. Furthermore, a high ESG evaluation can keep enterprises in a dominant position regarding their talent acquisition and value creation (Tang, 2022; Liu and Nemoto, 2021) [78,79]. Given the above, we propose Hypothesis 2:

**Hypothesis 2.** *ESG performance can promote enterprise value.*

Stakeholder theory holds that the development of any company cannot be separated from the input or participation of all stakeholders, and the enterprise must pursue the overall interests of stakeholders, rather than only the interests of limited subjects. Ecomark and ESG introduced in this paper are two methods for enterprises to disclose environmental information. Ecomark is mainly aimed at the general public. By satisfying consumers' preference for environmentally friendly products, it indirectly increases investors' investment desire and thus affects the enterprise value (from customers' perspectives). On the other hand, ESG directly faces enterprises and investors (from investor and stakeholders' perspectives). According to the conclusions of previous research, both of these environmental information disclosure methods affect enterprise value, so we can predict that there may be some relationship between the interaction of Ecomark and ESG regarding enterprise value. From the perspective of cost, the expense of obtaining an Ecomark is high, which places an economic burden on enterprises (Takeuch, 2011) [80], while the cost of ESG information disclosure is relatively low. Therefore, enterprises tend to focus on the use of ESG for environmental information disclosure. Thus, ESG has the potential to weaken the impact of Ecomark on enterprise value directly or indirectly. Given the above, we propose Hypothesis 3:

**Hypothesis 3.** *The promoting effect of Ecomark on enterprise value is affected by the ESG performance. With the increase in enterprise ESG performance and the time passage of ESG implementation, its crowd-out effect on Ecomark will become stronger and the influence of Ecomark on enterprise value will be weakened.*

## 4. Data and Model

### 4.1. Variable Definitions and Descriptive Statistics

The explained variable of this study is Tobin's Q, which represents the enterprise value. Tobin's Q is the ratio of assets' market value and its replacement value, which reflects the evaluation of enterprise value through the stock market and is commonly used to measure enterprise value in a large body of existing literature. We followed the research of Konar and Cohen (2001) [32] and Iwata and Okada (2011) [40] to calculate it as follows: Tobin's Q = (MVE +LTDEBT + CL + BVINV − CA)/TA. Here, MVE refers to the corporate stock market value, LTDEBT refers to long-term debt, CL refers to current debt, BVINV refers to the value of inventory assets, CA refers to the value of current assets and TA refers to total assets.

The core explanatory variable includes enterprise ESG performance and Ecomark. We used the ESG score from the Bloomberg database to measure enterprise ESG performance. The score is a composite assessment of a company's environmental, social and governance dimensions, ranging from 0.1 to 100, based on publicly available information for investors, such as corporate social responsibility reports, annual reports and websites. Ecomark is defined as a dummy variable—that is, if the enterprise has obtained Ecomark certification, it is 1, and it is 0 otherwise. With the gradual enhancement of the environmental awareness of enterprise stakeholders, obtaining Ecomark certification is believed to have the effect of reducing the environmental load and improving enterprise value.

Control variables that we integrated in this study were return on equity (ROE), return on assets (ROA), debt dependence (Debt), enterprise size (Size), sales growth rate (SalesG), operating profit growth rate (OPG) and market competition degree (Comp) with respect to the scholarly references (Nishitani, 2014 [16]; Yagi and Managi, 2013 [81]; Nakao et al., 2010 [35]) and the availability of data. Table 2 summarizes the variables and their specific definitions.

**Table 2.** Variable definition.

| Variable Name | Variable Code | Variable Definition | Data Source | Expected Sign |
|---|---|---|---|---|
| Enterprise Value | Tobin's Q | $\ln((MVE + LTDEBT + CL + BVINV - CA)/TA)$ | Development Bank of Japan/Japan Research Institute of Economy, Trade and Industry: JIP Database | |
| ESG Performance | ESG | ESG disclosure score in Bloomberg | Bloomberg Database | + |
| Ecomark | Ecomark | Obtained Ecomark Certification = 1 Otherwise = 0 | Weekly Toyo Keizai: CSR Enterprise Overview Survey Data | + |
| Return on Equity | ROE | Net profit/average net worth | | + |
| Return on Assets | ROA | Net profit/average total assets | | + |
| Debt Dependence | Debt | Liabilities/total assets | Weekly Toyo Keizai: CSR Enterprise Overview Survey Data Wind Database | - |
| Enterprise Size | Size | ln (total assets) | | + |
| Sales Growth Rate | SalesG | $(Sales_t - Sales_{t-1}/Sales_{t-1} \times 100\%$ | | + |
| Operating Profit Growth Rate | OPG | $(Op.profit_t - op.\ profit_{t-1})/op.\ profit_{t-1} \times 100\%$ | | + |
| Market Competition Degree | Comp | 1—(Op profit and loss—non op. expenses)/Sales | | + |

### 4.2. Sample and Data Sources

In the baseline model, we employed the enterprise Ecomark data from the Weekly Toyo Keizai: CSR Enterprise Overview Survey Dataset 2009. In this survey, among other aspects, all of Japan's listed and major non-listed companies were asked whether they had obtained Ecomark certification. We then matched and analyzed Japan's listed electrical equipment manufacturing enterprises' Ecomark data based on the survey results of the CSR Enterprise Overview 2015 to 2021. All enterprises' data were double-checked and confirmed from the official websites of listed companies.

In addition, the enterprises' ESG data were obtained from the Bloomberg database. The financial data from 2008 to 2009 were calculated using data from the Development Bank of Japan/Japan Research Institute of Economy, Trade and Industry: JIP Database. The financial data from 2015 to 2021 were calculated using the data in the Wind database. Other independent variables' data came from the data of Japan's listed electrical equipment manufacturing enterprises in the Weekly Toyo Keizai: CSR Enterprise Overview Survey Dataset. In this paper, the original data were processed as follows: 7 delisted enterprises and the enterprises with missing data from the electrical equipment manufacturing enterprises in the sample set of 2009 were excluded, 116 delisted enterprises and enterprises with missing variable observed values from 2015 to 2021 were deleted, and the data of enterprises with a small number of missing values were fixed by using the multiple omission method. Finally, we obtained a mixed cross-sectional dataset of 190 enterprises in 2009 and 63 enterprises from 2015 to 2021. The descriptive statistics of variables are reported in Table 3.

**Table 3.** Descriptive statistics.

| Variable | Obs. | Mean | Std. Dev. | Min. | Max. |
|---|---|---|---|---|---|
| Tobin's Q | 630 | −0.2992 | 0.7323 | −6.7643 | 1.9199 |
| ESG | 441 | 42.7900 | 11.4942 | 18.7607 | 69.5503 |
| Ecomark | 631 | 0.1680 | 0.3742 | 0 | 1 |
| OPG | 631 | −0.3795 | 14.5463 | −145.2359 | 308.7275 |
| SalesG | 631 | −0.0344 | 0.1828 | −0.9572 | 2.5397 |
| Comp | 631 | 1.1482 | 0.2193 | 0.2363 | 4.7322 |
| ROE | 631 | 13.2765 | 165.4224 | −464.0700 | 3973.4900 |
| ROA | 631 | 2.4927 | 5.9113 | −50.1750 | 23.1500 |
| Size | 441 | 9.5610 | 1.6761 | 4.6052 | 12.7682 |
| Debt | 631 | 0.1390 | 0.1729 | −1.4826 | 1.5530 |

*4.3. Empirical Model*

4.3.1. Baseline Regression Modeling

To test the impact of Ecomark on enterprise value, we constructed a cross-sectional OLS regression model, referring to Zhang and Zhao (2019) [82], as follows:

$$\text{Tobin's Q}_{i,t+1} = \alpha + \beta_1 \text{Ecomark}_{it} + \delta' X_{it} + \varepsilon_{it} \quad (1)$$

By adopting two-period cross-sectional data, we are able to capture the impact of Ecomark on enterprise value in different periods. In Equation (1), Tobin's $Q_{i,t+1}$ is the enterprise value of enterprise $i$ in the $t + 1$ period, and Ecomark$_{it}$ indicates whether enterprise $i$ has obtained Ecomark certification in the $t$ period. In order to control the missing data issue in the 2009 sample and manage the accuracy of the estimation, we exclude enterprise size from the control variables in the baseline model.

4.3.2. Moderating Effect Modeling

In order to study whether ESG promotes or crowds out Ecomark, we constructed a moderating effect model to examine the relationships between ESG, Ecomark and enterprise value. Since the variable Ecomark is a categorical variable and ESG is a continuous variable, the traditional grouped regression setting is not suitable. We then performed a hierarchical multiple regression model by introducing an interaction term, referring to Wen et al. (2005) [83] and Yang and Zhang (2017) [84], as follows:

$$\text{Tobin's Q}_{it} = \alpha + \beta_1 \text{Ecomark}_{it} + \beta_2 \text{ESG}_{i,t-1} + \beta_3 \text{Ecomark}_{it} * \text{ESG}_{i,t-1} \\ + \delta' X_{it} + \mu_i + \omega_t + \varepsilon_{it} \quad (2)$$

where Tobin's Q is the dependent variable of enterprise value; Ecomark indicates whether or not an enterprise has obtained Ecomark certification; and the ESG score represents enterprises' ESG performance with a lag of one period, in consideration of the assumption that ESG lags behind enterprise value. Considering that enterprises with higher enterprise value tend to have stronger environmental awareness, they are thus more likely to choose Ecomark. $X_{it}$ represents all control variables, and the selection of control variables is mainly based on three considerations: first, variables reflect the enterprises' operating capacity, including return on equity and return on total assets; second, considering that enterprise size differences may represent different capacities regarding their environmental responsibility, this study uses the natural log form of total assets as a variable to measure and control enterprise size differences; and third, because of the risk tolerance differences in different enterprises, debt dependency is taken as the control variable to approximately reflect enterprises' risk level. $M_i$, $\omega_t$ and $\varepsilon_{it}$ represent the individual fixed effect, year fixed effect and random error term, respectively. The focus of the moderating effect model is

the significance and size of the interaction coefficient $\beta_3$ between ESG and Ecomark. By calculating the first derivative of Ecomark in Equation (2), we obtain

$$\frac{\partial \text{Tobin's Q}_{it}}{\partial \text{Ecomark}_{it}} = \beta_1 + \beta_3 \text{ESG}_{i,t-1} \tag{3}$$

Equation (3) represents the total effect of Ecomark. If the empirical results of $\beta3$ are significant, it indicates that Ecomark's effect of promoting enterprise value is regulated by ESG. $\beta_3 > 0$ indicates that Ecomark promotes enterprise value with the increase in ESG. $\beta_3 < 0$, indicating that the effect of Ecomark promoting enterprise value decreases with the increase in ESG. If the empirical result is statistically significant and less than zero, then Hypothesis 3 is verified.

Since Ecomark, as the core variable, does not change over time, the first-order difference or demean estimation of fixed effects cannot be used. By referencing the method proposed by Amemiya et al. (1977) [85], a generalized least square (FGLS) estimation method was adopted to verify Equation (2). FGLS seeks to substitute the residual vector of each individual section into a sectional covariance matrix of heteroscedasticity and then obtain the parameter estimates using GLS. The FGLS estimation method can correct problems, such as heteroscedasticity, concurrent correlation and sequence correlation, caused by cross-sectional data, and improve the consistency and effectiveness of panel regression (Angris and Pischke, 2009) [86]. At the same time, considering the common impact of the enterprises' economic operation, such as a change in the overall economic situation, the time trend variable is added to the FGLS method to control the influence of this aspect.

## 5. Result Analysis and Discussion

### 5.1. Baseline Regression Result

The Ecomark system has been in place in Japan since 1989 and has been applied to a variety of products (and services). The Ecomark obtained by enterprises means that their products have less impact on the environment throughout their whole product life cycle (from production to disposal), which is conducive to environmental protection. Through its Ecomark system, Japan aims to build a sustainable society by enabling consumers to make environmentally friendly product choices and promoting the efforts of enterprises involved to improve the environment. In 2009, the Weekly Toyo Keizai: CSR Enterprise Overview Survey revealed whether listed enterprises in the electrical industry had obtained Ecomark certification. Therefore, this paper takes 2009 as the starting point to measure the impact of the Ecomark system on enterprise value before the implementation of the ESG system. In 2014, the Japan FSA first issued the Japan Stewardship Code, which mainly proposed seven principles for institutional investors and proxy advisers entrusted by institutional investors in regard to investing in the stocks of listed companies in Japan, requiring them to actively exercise shareholders' rights and engage in dialogue with the invested companies to contribute to their sustainable growth. Another important policy document is the Japan Corporate Governance Code, which was first published in 2015 by the FSA in conjunction with the Tokyo Stock Exchange. The Code stipulates that all listed companies should follow ESG principles from the perspective of corporate governance, and requires companies to pay more attention to stakeholders, cooperate with relevant parties on ESG issues and take appropriate measures to solve ESG problems. The Code strengthens the role of the board by incorporating sustainable development issues and ESG elements into the board's responsibilities and obliging the board to proactively address and actively act on these matters. Considering that the impact of ESG information disclosure on enterprise value has a certain time lag, we take 2021 as the second key time point to explore the evolution of Ecomark's impact on enterprise value against the background of the continuous development of ESG.

Table 4 presents the estimation results of Ecomark in different time periods. According to the regression results of Model 1, Ecomark has a significant positive correlation with enterprise value at a level of 1%, indicating that the Ecomark certification of Japan's listed

electrical equipment manufacturing enterprises has a tendency to promote the improvement of enterprise value. This is consistent with the conclusions of Nishitani and Kokubu (2012) [37] and Aguilar and Cai (2010) [48]. In other words, enterprises tend to increase their financial performance by improving their environmental performance. Therefore, Hypothesis 1 is confirmed. Based on the comparison between Model 1 and Model 2, there is a significant difference between 2009 and 2021. Specifically, the value of Ecomark to the enterprise changes from positive significance at the 1% level to negative significance at the 5% level. On one hand, this may be due to policy factors whereby enterprises prefer the ESG performance evaluation system that directly faces investors; on the other hand, it may be due to the fact that the cost of the Ecomark is much higher than the cost of ESG information disclosure. Therefore, one possible explanation is that with the continuous improvement in ESG and the passage of time, the role of the Ecomark system in enterprise value may tend toward to being crowded out by ESG.

**Table 4.** Empirical results of baseline regression.

| Variables | Model 1 (2009) Tobin's Q | Model 2 (2021) Tobin's Q |
|---|---|---|
| Ecomark | 0.4661 *** | −0.3902 ** |
| | (4.22) | (−2.19) |
| OPG | 0.0006 | 0.0232 |
| | (0.18) | (0.86) |
| SalesG | 0.0348 | 2.2241 ** |
| | (0.18) | (3.14) |
| Comp | 0.3042 | 0.9831 |
| | (1.75) | (1.35) |
| ROE | 0.0702 | −0.0101 |
| | (1.23) | (−0.84) |
| ROA | 0.6992 | 0.0479 |
| | (0.89) | (1.65) |
| Debt | 1.0001 * | −0.0911 |
| | (2.38) | (−0.14) |
| _cons | −1.0132 *** | −1.4573 |
| | (−5.29) | (−1.68) |
| Sample Size | 190 | 63 |
| R-squared | 0.0793 | 0.3418 |

Note: *** $p < 0.01$; ** $p < 0.05$; * $p < 0.1$. The value in parentheses below the coefficient is the standard error.

### 5.2. Regression Results of Moderating Effect Model

From Model 3's regression results, it can be seen that ESG performance is significantly positively correlated with enterprise value at the 5% level, indicating that it has a positive impact on enterprise value without controlling the year fixed effect. Model 4 controls the bidirectional fixed effect, and the overall regression result has little deviation from Model 3. At this time, the ESG coefficient is also significantly positive and its coefficient value increases when compared with Model 3, which is consistent with the research conclusions of Cajias et al. (2014) [52] and Wong et al. (2021) [54]. Thus, Hypothesis 2 is verified.

Models 5 and 6 add further interaction terms between Ecomark and ESG to capture the moderating effect of ESG on Ecomark. At this time, the enterprise ESG coefficient is still significantly positive at the level of 1%, and the significance level and coefficient are slightly higher than those of Model 4. The Ecomark coefficient is significantly positive at the 5% level, which verifies the conclusions of Model 1. The interaction coefficient is significantly negative at the 5% level, which is in line with expectations (results shown in Table 5). Based on the above analysis, under the influence of ESG, the Ecomark impact on

enterprise value is to calculate the reciprocal of the first derivative of Ecomark in Equation (3), which is the following equation:

$$\text{Ecomark} = 1 / \frac{\partial \text{Tobin'sQ}_{it}}{\partial \text{ESG}_{it}} \tag{4}$$

**Table 5.** Regression results of moderating effect model.

| Variables | Model 3 Tobin's Q | Model 4 Tobin's Q | Model 5 Tobin's Q | Model 6 Tobin's Q |
|---|---|---|---|---|
| ESG | 0.0111 ** | 0.0117 ** | 0.0155 *** | 0.0154 *** |
| | (2.51) | (2.41) | (3.35) | (3.11) |
| Ecomark | | | 1.2271 ** | 1.2221 ** |
| | | | (2.1) | (2.14) |
| ESG * Ecomark | | | −0.0282 ** | −0.0279 *** |
| | | | (−2.54) | (−2.56) |
| OPG | −0.0053 | −0.0072 | −0.0046 | −0.0065 |
| | (−1.14) | (−1.58) | (−1.00) | (−1.45) |
| SalesG | 1.4023 *** | 1.6162 *** | 1.3611 *** | 1.5690 *** |
| | (5.25) | (5.67) | (5.16) | (5.57) |
| Comp | 0.2491 | 0.2032 | 0.5531 | 0.5125 |
| | (0.74) | (0.6) | (1.56) | (1.45) |
| ROE | 0.0003 | 0.0003 | 0.0001 | 0.0001 |
| | (0.44) | (0.42) | (0.1) | (0.11) |
| ROA | 0.0271 *** | 0.0285 *** | 0.0269 *** | 0.0286 *** |
| | (4.12) | (4.38) | (4.14) | (4.43) |
| Size | −0.0591 * | −0.0614 * | −0.0436 | −0.0439 |
| | (−1.95) | (−1.92) | (−1.34) | (−1.30) |
| Debt | −0.1841 | −0.2114 | −0.1273 | −0.1532 |
| | (−0.96) | (−1.11) | (−0.66) | (−0.82) |
| _cons | −0.4821 | −0.3135 | −1.1292 ** | −0.9796 |
| | (−1.01) | (−0.64) | (−2.12) | (−1.79) |
| Year Fixed | NO | YES | NO | YES |
| N | 378 | 378 | 378 | 378 |

Note: *** $p < 0.01$; ** $p < 0.05$; * $p < 0.1$. The value in parentheses below the coefficient is its standard error.

From the above equation, it can be seen that the total effect of Ecomark on enterprise value is regulated by ESG. Since the influence of ESG continuously accumulates and increases, the total effect of Ecomark on enterprise value decreases over time, and Hypothesis 3 is verified. Moreover, the empirical results show that the total effect of Ecomark on enterprise value will decrease by 0.0279 units per unit increase in ESG.

Regarding the control variables, the sales growth rate has a positive impact on enterprise value at the 1% level of significance. This is because the stronger the growth ability of enterprises, the better the investors' expectation regarding the development prospects of enterprises. Therefore, the enterprise value will be significantly improved (Elsayed and Paton, 2005 [34]; Nishitani and Kokubu, 2012 [37]). There is a significant negative correlation between enterprise size and enterprise value at the 10% level, which may be because the measurement of Tobin's Q is the ratio of enterprise market value to total assets. For enterprises of a large scale, the growth of the profitability of assets may be lower than the expansion speed of the asset scale. Thus, the growth in enterprise size decreases enterprise value (Nishitani, 2014) [16]. Debt dependency has a negative impact on enterprise value. Although it does not pass the significance test, it has the same tendency as the analysis results of Iwata and Okada (2011) [40]. This may be because the increased risks in the operation process of the enterprise are unpredictable. According to the theory of capital structure equilibrium, the higher the debt–equity ratio, the faster the value of the enterprise will decline.

### 5.3. Robustness Check

5.3.1. Model Specification Check

When the time number T is less than the section number N in the panel data model, the standard deviation of the FGLS estimation parameter cannot fully reflect its variation, so the panel correction standard error (PCSE) should be considered to deal with the complex panel error structure. On the basis of retaining the OLS estimation parameters, the PCSE method replaces the residual term with the diagonal matrix to correct its standard deviation, which is an alternative to FGLS and can perform the regression estimation of panel data more accurately (Beck and Katz, 1995) [87]. Therefore, in order to further improve the robustness of the estimation results, we use the PCSE measurement method to estimate Equation (2). The estimation results are shown as Model 7 in Table 6. The interaction coefficient between ESG and Ecomark is still significantly positive; thus, the conclusion is robust and is not affected by the time interval and estimation method.

**Table 6.** Results of robustness check/endogeneity test.

| Variables | Model 7 PCSE | Model 8 Winsorize | Model 9 System GMM |
|---|---|---|---|
| Tobin's $Q_{t-1}$ | | | 0.5542 * |
| | | | (1.82) |
| Ecomark | 1.4571 *** | 1.2442 ** | 1.0391 ** |
| | (4.38) | (2.49) | (2.03) |
| ESG | 0.0202 *** | 0.0153 *** | 0.0190 ** |
| | (3.95) | (3.51) | (2.49) |
| ESG*Ecomark | −0.0325 *** | −0.0292 ** | −0.0245 ** |
| | (−4.82) | (−3.07) | (−2.17) |
| OPG | −0.0037 | −0.0051 | 0.0029 |
| | (−0.70) | (−1.28) | (1.08) |
| SalesG | 0.6312 * | 1.3001 *** | 0.5963 ** |
| | (1.78) | (5.28) | (2.04) |
| Comp | 0.2982 | 0.5301 * | 0.8182 |
| | (0.93) | (1.72) | (0.74) |
| ROE | 0.0001 | −0.0001 | 0.0002 |
| | (0.25) | (−0.14) | (0.26) |
| ROA | 0.0109 | 0.0245 *** | 0.0103 |
| | (1.47) | (4.34) | (0.70) |
| Size | −0.0782 ** | −0.0310 | −0.0683 |
| | (−2.43) | (−1.05) | (−0.59) |
| Debt | 0.0601 | −0.1801 | 0.1152 |
| | (0.29) | (−1.09) | (0.40) |
| _cons | −0.6044 | −1.1032 * | −1.2072 |
| | (−1.23) | (−2.31) | (−0.58) |
| Year Fixed | YES | YES | YES |
| N | 378 | 378 | 378 |
| AR(1) | | | −2.65 |
| *p*-value | | | (0.008) |
| AR(2) | | | 1.59 |
| *p*-value | | | (0.113) |
| Hansen Test | | | 52.49 |
| | | | (0.206) |

Note: *** $p < 0.01$; ** $p < 0.05$; * $p < 0.1$. The value in parentheses below the coefficient is the standard error.

5.3.2. Eliminating Outlier Interference

The outliers of explained variables will have a certain impact on the regression results. In order to reduce the interference of outliers, in this paper, a winsorize process was performed on the 1% and 99% percentiles of enterprise value, and the estimated results are shown as Model 8 in Table 6. The regression results after the winsorization process were

consistent with the regression results given above. This shows that the conclusion is robust and not affected by sample fluctuations.

### 5.4. Endogeneity Test

Considering that Ecomark may have a bidirectional causal relationship with enterprise value in terms of the decision of whether to use Ecomark, resulting in the endogeneity problem, we included the first-order lag term of the dependent variable as an explanatory variable in the empirical model and conducted an endogeneity test by constructing a dynamic panel data equation (Equation (5)). The system-generalized moment estimation method (system GMM) can estimate the variables that do not change with time, and the estimation efficiency is higher. Even if the variables have measurement errors, the consistent estimation can still be obtained. Since the core variable (Ecomark) does not change with time, the system GMM method is more appropriate for estimation (Bond et al., 2001) [88]. Therefore, we take the first-order difference lag term of the explained variable as the instrumental variable and use the system GMM method to estimate Equation (5) with respect to the same control variable group in Equation (2). The estimations of AR(1) and AR(2) are reported as Model 9 in Table 6. The results show that the *p*-value of first-order autocorrelation is less than 0.05, while the *p*-value of second-order autocorrelation is greater than 0.1, indicating that the residual term of Equation (5) has a significant first-order sequence correlation without second-order autocorrelation, which meets the moment constraint of the GMM estimation of the system. The result of Hansen's test confirms that the instrumental variables used by the model are effective. According to the results of Model 9, the first-order lag term of enterprise value positively influences the enterprise value of the current period at the significance level of 10%. ESG has a crowd-out effect on Ecomark. ESG and Ecomark are helpful in the promotion of the growth of enterprise value, which further confirms the above research conclusions. Except for certain variables, the regression results of control variables are consistent with the above results, which support the reliability.

$$
\begin{aligned}
\text{Tobin's Q}_{it} &= \alpha + \beta_1 \text{Tobin's Q}_{i,t-1} + \beta_2 \text{Ecomark}_{it} + \beta_3 \text{ESG}_{i,t-1} \\
&+ \beta_4 \text{Ecomark}_{it} * \text{ESG}_{i,t-1} + \delta' X_{it} + \mu_i + \omega_t + \varepsilon_{it}
\end{aligned}
\tag{5}
$$

## 6. Conclusions

In recent years, much more attention has been paid to enterprises' performance and activities regarding the environment. Many studies have verified the link between enterprise environmental information disclosure and enterprise value. However, most of the existing literature merely considers the impact of a certain environmental information disclosure method on enterprise value, without taking into account the whole product life cycle, which has left a research gap and allowed us to compare different environmental information disclosure methods in enterprise value. Compared to previous studies, this paper analyzes a sample of Japan's listed electrical equipment manufacturing enterprises and provides strong evidence to demonstrate how different environmental information disclosure methods impact enterprise value. We confirm that Ecomark and ESG have a tendency to promote enterprise value, which is consistent with the research conclusions of most scholars. Due to the differences in time, cost and environmental information disclosure method, this paper incorporates Ecomark and ESG environmental information disclosure methods into a unified analysis system to study the impact on enterprise value. To quantify the link and interaction between Ecomark and ESG regarding enterprise value, this paper further employs a series of empirical models and robustness checks. The empirical results show that the interaction term is negatively correlated with Tobin's Q, which means that, along with the passage of time, ESG reduces the promoting effect of Ecomark on enterprise value. This implies the ESG has a crowding-out effect of Ecomark on enterprise value. A possible explanation may associated with the enterprises' low-cost driving preferences and the potential investment opportunities that stakeholders contributed. However, whether

the crowding-out effect has industrial universality is still an unresolved question and one of the limitations of this paper, which is worth discussing in a series of further studies.

Undoubtedly, good and low-cost ESG performance will be more conducive to the promotion of enterprise value. From a practical perspective, the main findings support that an ESG system that directly affects investors may have a stronger impact on enterprise value than the Ecomark system's indirect influence through consumers. The possible implication for enterprise decision-makers is that the expansion of environmental information disclosure methods may optimize enterprise value.

Moreover, the environmental information disclosure policy guides enterprises to better fulfill their environmental and social responsibilities by influencing their value. Driven by the value-maximizing goal, enterprise management tends to cater to investors' preferences regarding environmental social responsibility. However, investors' preferences for environmental social responsibility depend on the existing environmental information disclosure indicators of the enterprise, which are mainly represented by ESG. Compared with the Ecomark system, the ESG system lacks the environmental monitoring of the product life cycle and ignores the consumers' social value in attaining an environmentally friendly lifestyle. This may cause issues in terms of integrating behavioral externality into the ESG mechanism. Therefore, establishing a suitable environmental information disclosure framework to guide enterprise investors' behavior by coupling the Ecomark system and the ESG system may be the best method towards the sustainable development of enterprises and society.

**Author Contributions:** Conceptualization, A.G. and Y.L.; methodology, D.M. and T.X.; validation, T.X., Y.L. and D.M.; formal analysis, A.G.; data curation, A.G. and T.X.; writing—original draft preparation, A.G. and T.X.; writing—review and editing, Y.L. and D.M.; funding acquisition, A.G. All authors have read and agreed to the published version of the manuscript.

**Funding:** This research was funded by a Guangxi Normal University Doctor's Research Start-Up Grant (A-402-00-0055A4) and the Guangxi Philosophy and Social Sciences Planning Research Project (20BJY001).

**Institutional Review Board Statement:** Not applicable.

**Informed Consent Statement:** Not applicable.

**Data Availability Statement:** The data that are presented in this study are available within the tables. They are also available from the corresponding author upon request.

**Conflicts of Interest:** The authors declare no conflict of interest.

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
