# Peer review of "Promote or Crowd Out? The Impact of Environmental Information Disclosure Methods on Enterprise Value"

_sustainability, doi:10.3390/su15043090_

Round 1

Reviewer 1 Report

This article examines the traditional methods of disclosing environmental information (such as Ecomark) and their impact on company value. The data was collected from Japanese electrical equipment manufacturing companies. In my opinion, this article in its current version is not suitable for publication. The problems are as follows:
1. The terms of data collection are unclear to me. Authors must publish the data and specify exactly how they obtained the data.
2. In my opinion, the English of this article is crowded, jumbled, and sloppy. it is unacceptable.
3. The authors claim that Ecomark acquisition and good ESG performance can help improve company value. I don't see what supports this claim. Also, what is the advantage and importance of this finding? If this finding has no practical benefit, why should a research be defined for it?
4. The analysis method and model of this article is simple and full of flaws. Authors should describe the model precisely. Explain why they chose this model. In my opinion, the current model is not acceptable and does not meet the criteria of a research paper.

Author Response

Dear Reviewer,

We are grateful to your valuable comments and suggestions. Based on your reviewing report, we did a point-by-point rechecks and revisions. A detailed response is attached. We hope that our revised manuscript and response can make you satisfied.

Again, thanks for your time and comments. We look forward to hearing from you.

Sincerely,

Anrong Gao, Tianren Xiong, Yuxi Luo and Defeng Meng

Reviewer 2 Report

Thank you very much for the opportunity to review the manuscript. 

The manuscript is well-written and organized. However, I have some suggestions to be revised before publishing the manuscript as follows:

In the Literature Review section, please add more related research papers in the context of Japan and Japanese firms in general. 

Also, there were no related research papers in the context of Japanese firms in  Hypothesis 1 to 3.

Moreover, there were no related papers about the moderating effect of  ESG and crowd-out Ecomark. Please add more relevant research papers.

Last but not least, a proofreading service is suggested. There is no incomplete sentence. For example,  "in order to study whether ESG promote or crowd-out Ecomark, a moderating effect model was established."

Finally, please add managerial and theoretical implications.

Author Response

(The authors gave the same response as above.)

Reviewer 3 Report

some amendments in the format are due to admit publication. The main problem is the epistemological structure (why the article was conceived and how the study was developed). I suggest the following structure of objectives: (i) research gap; (ii) research question; (iii) purpose of the article; (iv) intermediate objectives; (v) assumptions or hypo; and (vi) research method. This structure must appear in the introduction.

The research gap must be created by a systematic literature review that provides 'holes' in the state of knowledge on the topic. I believe that a full review should not be done, but an analysis of about 5-8 studies on the topic under discussion. You can find some examples, which will show the relevance of the issue, as it is indeed a topic of current, relevant research. At the end of the justification you should write something like: According to what we were able to find, there are no studies referring and reporting on ... With this you have therefore proven that the issue is relevant, and you have also proven that your study does indeed fill a research gap.

I propose hypotheses to be presented in the chapter research methodology (here Data). The literature review should include what has been written on the topic.

In conclusion, I propose

-evaluate the critical research, show its limitations and weaknesses,

- highlight the new knowledge and the lessons learned from it,

- describe the importance of the research and how it affects the wider field, show how the information obtained can be further used

Conclusions must be clearly and unambiguously linked to the results of the survey. Their theoretical and practical implications should be indicated (This section is not mandatory but can be added to the manuscript if the discussion is unusually long or complex.)

Author Response

(The authors gave the same response as above.)

Reviewer 4 Report

Please take into consideration the notes in the file attached.

Please consider the following references where pointed in the notes attached:

1. Ruiz-Blanco, S., Romero, S. & Fernandez-Feijoo, B. (2022). Green, blue or black, but washing - What company characteristics determine greenwashing? Environment, Development and Sustainability, 24. 4024-4045.

2. Tettamanzi, P., Venturini, G. & Murgolo, M. (2022). Sustainability and Financial Accounting: a Critical Review on the ESG Dynamics. Environmental Science & Pollution Research. 29, 16758-16761.

3. Zhong, S., Hou, J., Li, J. & GAo, W. (2022). Exploring the Relationship of ESG Score and Firm Value using fsQCA Method: Cases of the Chinese Manufacturing Enterprises. Frontiers in Psychology, 13.

4. Yue, Q. & Wang, Y. (2022). Research on the Impact of ESG Information Disclosure on the Value of Resource-Based Enterprises - An Analysis of the Mediating Effect Based on Enterprise Risk-Taking. Proceedings of the 9th Academic Conference of Geology Resource Management and Sustainable Development, 1226-1230.

Author Response

(The authors gave the same response as above.)

Round 2

Reviewer 1 Report

I think the paper can be published in this version.

Author Response

Dear Reviewer,

We are grateful to see your affirmative decision. Thank you very much for your help and supports. We will keep polishing our manuscript for publishing.

With regards,

Anrong Gao, Tianren Xiong, Yuxi Luo and Defeng Meng 

Reviewer 4 Report

The paper would benefit from a linguist review for clarity and consistency. Moreover, the abstract still lacks of information pertaining to the methodology applied and the sample chosen as well as to a clear definition of the research objectives.

Author Response

Dear Reviewer,

We are grateful to your valuable comments and suggestions. Based on your reviewing report, we modified the abstract with including research objectives and methods and rephrasing main findings. We also did a language edits for the our manuscript. We hope that our revised manuscript and response can make you satisfied.

Again, thanks for your time and comments. We look forward to hearing from you.

Sincerely,

Anrong Gao, Tianren Xiong, Yuxi Luo and Defeng Meng
